# PARP-1/2 Inhibitor Olaparib Prevents or Partially Reverts EMT Induced by TGF-β in NMuMG Cells

**DOI:** 10.3390/ijms20030518

**Published:** 2019-01-26

**Authors:** Michelle Schacke, Janani Kumar, Nicholas Colwell, Kole Hermanson, Gustavo A. Folle, Sergei Nechaev, Archana Dhasarathy, Laura Lafon-Hughes

**Affiliations:** 1Instituto de Investigaciones Biológicas Clemente Estable, Montevideo 11600, Uruguay; asistentes@iibce.edu.uy (M.S.); gfolle@iibce.edu.uy (G.A.F.); 2Department of Biomedical Sciences, School of Medicine and Health Sciences, University of North Dakota, Grand Forks, ND 58202-9061, USA; kumarjanani17@gmail.com (J.K.); nicholas.colwell@und.edu (N.C.); kole.hermanson@und.edu (K.H.); sergei.nechaev@und.edu (S.N.)

**Keywords:** EMT, actin cytoskeleton, anisotropy, Poly(ADP-ribose), PARP, PAR belt, olaparib, XAV939, MEO328

## Abstract

Poly- adenosine diphosphate (ADP)-ribose (PAR) is a polymer synthesized as a posttranslational modification by some poly (ADP-ribose) polymerases (PARPs), namely PARP-1, PARP-2, tankyrase-1, and tankyrase-2 (TNKS-1/2). PARP-1 is nuclear and has also been detected in extracellular vesicles. PARP-2 and TNKS-1/2 are distributed in nuclei and cytoplasm. PARP or PAR alterations have been described in tumors, and in particular by influencing the Epithelial- Mesenchymal Transition (EMT), which influences cell migration and drug resistance in cancer cells. Pro-EMT and anti-EMT effects of PARP-1 have been reported while whether PAR changes occur specifically during EMT is currently unknown. The PARP-1/2 inhibitor Olaparib (OLA) is approved by FDA to treat certain patients harboring cancers with impaired homologous recombination. Here, we studied PAR changes and OLA effects on EMT. Total and nuclear PAR increased in EMT while PAR belts were disassembled. OLA prevented EMT, according to: (i) molecular markers evaluated by immuno-cytofluorescence/image quantification, Western blots, and RNA quantitation, (ii) morphological changes expressed as anisotropy, and (iii) migration capacity in the scratch assay. OLA also partially reversed EMT. OLA might work through unconventional mechanisms of action (different from synthetic lethality), even in non-BRCA (breast cancer 1 gene) mutated cancers.

## 1. Introduction

Poly-ADP-ribosylation (PARylation), which is a post-translational modification, consists of up to 400 linked ADP-ribose residues constituting a poly-ADP-ribose (PAR) linear or branched chain. PAR is synthesized from Nicotinamide adenine dinucleotide (NAD+) by the poly-ADP-ribose polymerase (PARP) enzymes, which catalyze the transfer of ADP-ribose to target proteins. The PAR chains on these target proteins can interact non-covalently with specific protein domains, namely PBZ, WWE, PBM, RRM, and macro-domains. PAR is degraded mainly by poly-(ADP-ribosyl) glycohydrolase (PARG) activity [1,2,3]. 

PARylation and/or PARP expression is altered in several pathologic contexts [4,5,6,7,8], including cancers of epithelial origin. The molecular mechanisms of such alterations have been interpreted mainly in terms of chromatin structure modulation or sensitization to DNA damage [5,9,10,11]. The PARP-1/2 inhibitor Olaparib (OLA, Lynparza, AZD-2281) has been approved by U.S. Food and Drug Administration (FDA) to treat certain cancers, especially in BRCA-mutated patients (ucm572143, ucm592357, and NCT02987543). The rationale for designing clinical trials was the synthetic lethality expected when *brca* mutant patients were treated with OLA [9,10].

PARylation biology is quite complex and still poorly understood. The PARP family has 18 members [12], four of which have PARylating activity. PARP-1 and PARP-2 synthesize long branched PAR [13], as shown by Atomic Force Microscopy (AFM) [14], whereas Tankyrase-1 and Tankyrase-2 (TNKS-1/2) synthesize short, linear PAR. PARP-9 and PARP-13 have no detectable activity. All other PARPs, including PARP-3, accomplish mono-ADP-ribosylation [2,3,13,15,16]. The archetypal PARP-1 shows an exclusively nuclear localization [17]. Accordingly, most studies are focused on nuclear PARylation. There is a nuclear basal pool and another pool that is induced by genotoxic stress. PARP inhibitors (PARPis) increase the sensitivity to induced genotoxic damage [18,19,20]. The PAR scientific community agrees that nuclear PARPs affect chromatin remodeling, transcription, DNA replication, DNA repair, telomeric length regulation, and cell cycle control [21]. 

Cytoplasmic PAR roles are much less studied in spite of the fact that most PARPs, including PARP-2, TNKS-1/2, and PARP-3, can be found both in nuclei and cytoplasm [17]. TNKS-1 transiently associates with epithelial cell junctions [22] and a PAR belt exists in E-cadherin-rich epithelia, which was not detected in N-cadherin-rich bovine cornea cells. The PAR belt is a ring of only 1.5 µm in height that surrounds each epithelial cell running just below the tight junctions, encircling each of the interacting cells in the sheet. Its name recalls its similarity in position and apparent dimensions to the epithelial adhesion belt (or *zonula adherens*), as seen under a confocal microscope through the labeling of any of its components (e.g., E-cadherin). PAR distribution changes are observed in close correlation with F-actin or E-cadherin disturbances both in epithelia and in mouse sciatic nerves, which suggests that, in mammalian cells, there is a PAR pool structurally and functionally related to E-cadherin-rich cell junctions and the F-actin cytoskeleton [8,23]. 

A key feature of cancer progression is the epithelial to mesenchymal transition (EMT), which is a process of trans-differentiation of epithelial to mesenchymal cells. This is thought to contribute to both cancer cell migration and to chemo-resistance [24,25,26,27,28]. During EMT, the delocalized and diminished epithelial markers (E-cadherin, ZO-1) are replaced with mesenchymal markers (N-cadherin, vimentin). Furthermore, the actin cytoskeleton undergoes remodeling: the epithelial cortical F-actin ring (originally anchored by linking proteins to E-cadherin rich junctions) is disassembled, while conspicuous F-actin stress fibers are built and the cell undergoes flattening and spreading [27,28]. Thus, an EMT model seemed ideal to test the existence of a structural and functional relationship between cytoplasmic PAR, E-cadherin junctions, and the F-actin cytoskeleton. 

EMT is also characterized by nuclear accumulation of β-catenin, Smad-2/3, NF-κB (nuclear factor kappa-light-chain-enhancer of activated B cells), Snail, and Slug [24,29]. About 90% of cancers in humans originate from epithelial cells [30] and molecular changes consistent with EMT have been detected in many of them [31,32,33].

EMT can be triggered by inflammatory signals such as transforming growth factor beta (TGF-β), oxidative stress, or mechanical injury as well as by changes in pH or metabolism [34]. The underlying TGF-β induced cell signaling network is quite complex, involving transcription changes of 10% of analyzed genes [35] through canonical and non-canonical pathways [36,37,38].

Additionally, TGF-β was shown to induce a sustained PARP-3 increase from 3 to 72 h of treatment initiation. PARP-3 expression correlates with the expression of the mesenchymal marker vimentin while being in inverse correlation with E-cadherin in various cell lines. The same group also demonstrated that PARP-3 silencing has the effect of blocking migration and etoposide resistance induced by TGF-β [39]. PARP-1 also participates in the inflammatory response [40]. Both a pro-EMT [41] and an anti-EMT [42,43] PARP-1 role have been proposed.

We surmised that, if PAR changes in EMT effectively occurred, then it would be worth assessing the effect of PARP inhibitors (PARPis). We chose NAMRU Murine Mammary Gland (NMuMG) cells, which were the first model of EMT induction using TGF-β [27,28] and remained among the best *in vitro* EMT models. We measured typical changes in molecular markers E-cadherin or β-catenin and vimentin. We also wanted to quantify the extent of morphological changes including nuclear shape and F-actin reorganization. Anisotropy (opposed to isotropy) is the quality of exhibiting physical or mechanical properties (absorbance, elasticity, temperature, and conductivity) with different values when measured along axes in different directions. Anisotropy is most easily observed in single crystals of solid elements or compounds, in which atoms, ions, or molecules are arranged in regular lattices. In contrast, the random distribution of particles in liquids, and especially in gases, causes them rarely, if ever, to be anisotropic (see figshare online digital data repository link for anisotropy information and examples, doi 10.6084/m9.figshare.7505327). Based on the anisotropy concept, we quantified the orientation and alignment degree of the nuclei or the fibrillar F-actin filaments. Lastly, migration capacity was assessed through scratch assays. PARP-1/2 inhibitor Olaparib, like the PARP-3 inhibitor MEO328 (MEO) and unlike the tankyrase inhibitor XAV939 (XAV), hampered or reversed EMT induced by TGF-β in NMuMG cells. Refining the molecular mechanisms involved is beyond the scope of this work. Our results argue in favor of a pro-EMT role of PARP-1/2 in this system although off-target Olaparib effects cannot be discarded. In any case, as NMuMG cells express *BRCA* genes performing functions consistent with normal genes [44] and a BRCA mutation has not been reported in NMuMG cells, our results suggest that the Olaparib scope of action may be wider than in BRCA-mutated cells and might be beyond synthetic lethality, which is encouraging.

## 2. Results

### 2.1. EMT Induced Total and Nuclear PAR Increase as well as PAR Belt Disassembly

We wanted to test whether E-cadherin rich cells harbored a PAR belt as well as if there were changes in this belt and in nuclear/cytoplasmic PAR pools during TGF-β-induced EMT. NMuMG cells were exposed to TGF-β for 48 h and compared to control non-treated cells. A second control consisted of co-treatment with SB-431542, which is a TGF-β inhibitor, for visual assessment in order to confirm that the observed TGF-β effect depended on the serine/threonine kinase activity of type I receptors [45]. When SB-431542 was dissolved in 0.2% Dimethyl sulfoxide (DMSO), vehicle controls (0.2% DMSO in the absence or presence of TGF-β) were also carried out (Figure 1). We used confocal microscopy (see methods) to obtain both low-resolution (Figure 1A–I) and high-resolution (Figure 1J–R) images. 

We observed a clear and distinct PAR belt in untreated NMuMG cells (Figure 1A,D,G,J,M,P). Upon TGF-β treatment (Figure 1B,E,H,K,N,Q), the PAR belt and the cortical actin ring were disassembled. Total PAR increase was more pronounced than nuclear PAR increase, implying that cytoplasmic PAR increased while conspicuous stress fibers were assembled. All these changes were mitigated by the addition of SB-431542 diluted in DMSO (Figure 1C,F,I,L,O,R). No mitigation was observed in the presence of DMSO alone (10.6084/m9.figshare.7467530, 10.6084/m9.figshare.7492961). Apart from PAR detection with two different anti-PAR antibodies, experiments were performed to confirm PAR identity. XAV939 TNKS inhibitor blocked PAR belt synthesis (Figure A1, top panel) and digestion of PAR by PARG in fixed cell preparation abolished the signal (Figure A1, bottom panel). This is consistent with our previous findings in green monkey renal epithelial VERO cells [23]. After TGF-β treatment, the actin belt and the PAR belt were disassembled (Figure 1B,E,H,K,N,Q).

Next, we measured total and nuclear PAR differences using semi-quantitative measurements on immunocytofluorescence (ICF) images. Both total and nuclear PAR staining increased significantly (Figure 1S–T). The Image J Intensity Ratio Nuclei cytoplasm macro (IRNCM) segments the signal in a channel of interest (in this case, the green channel depicting PAR signal), into nuclear or cytoplasmic, using the DAPI (4′,6-diamidino-2-phenylindole) channel as a spatial reference to delimitate nuclear borders. We believe that the nuclear signal measurements were correctly performed, as long as no saturation was present. Cytoplasmic signal measurements were not adequately performed by the IRNCM in our system. This was probably due to the absence of a proper cytoplasmic or membrane marker to establish boundaries in cytoplasmic areas from cell-free background areas, which was combined with dramatic changes in cell size and shape that occur during EMT. Nevertheless, the increases both in total and nuclear PAR indicated that cytoplasmic PAR increased as well. This PAR increase stimulated our interest in testing the effect of PARP inhibitors in this system.

### 2.2. Anisotropy Measurements of Cell Morphology Changes

Cell biology heavily relies on the behavior of fibrillar structures, such as the cytoskeleton, yet the analysis of their behavior in tissues often remains qualitative. FibrilTool ImageJ macro (FTIJ), based on the concept of nematic tensor, provides a quantitative description of the anisotropy of fiber arrays and their average orientation in cells. It has already been validated on actin micro-fibrils [46].

Following EMT induced by TGF-β, as expected, the actin cytoskeleton distribution changed. Instead of the actin rings observed in control cells (Figure 1G or Figure 1P), there were prominent stress fibers arranged in a predominant direction beyond each individual cell, which generates a fiber alignment throughout the cell population (Figure 1H or Figure 1Q). Nuclei were not only bigger but also more ovoid, with their major axis running in parallel to the stress fibers. This ordered alignment was reflected in the anisotropy vectors. For example, F-actin anisotropy was represented in a certain scale, which was maintained throughout the measurements in each experiment, as green bars superimposed onto the actin channel (Figure 1G–I,P–R). Using FTIJ, anisotropy was measured in the F-actin (Figure 1U) and DAPI (Figure 1V) channels, which shows statistically significant anisotropy increases. After this initial quantification, the F-actin channel, which showed a much larger anisotropy change, was selected for further measurements. Then a strict anisotropy measurement criterion was established by using only the three planes from each confocal stack depicting higher F-actin total signal in whole-field intensity measurements. Accurate anisotropy measurements were incorporated in our experimental design as a way of quantitatively expressing the effects of OLA in this system.

### 2.3. Treatment with OLA Prevented TGF-β Induced EMT in NMuMG Cells

To determine whether OLA could affect TGF-β induced EMT, we performed co-treatment of NMuMG cells with OLA and TGF-β (48 h). After ICF and confocal microscopy, we performed a semi-quantitative evaluation of changes in the abundance of E-cadherin (Figure 2A–C,Q), vimentin (Figure 2G–I,R) as well as PAR (Figure 2T). F-actin anisotropy was also assessed. A low OLA concentration (50 nM), which tended to diminish PAR (Figure 2T), notably counteracted TGF-β-induced EMT-associated changes in vimentin (Figure 2R) and E-cadherin levels (Figure 2Q) as well as in anisotropy (Figure 2S). 

We next evaluated the effects of co-treatment with TGF-β and each of three different PARP inhibitors at the same concentration (10 µM). To do this, we pre-treated serum starved cells with the PARP inhibitors for 1 h, which was followed by TGF-β treatment for a further 48 h (PARP inhibitors were not removed). EMT changes were evaluated through E-cadherin RT-PCR (Figure 3B) and Western blots (Figure 3C). As expected, we noticed statistically significant downregulation of both RNA and protein E-cadherin levels with TGF-β treatment. Treatment with XAV did not interfere with this loss. However, both MEO and OLA showed no difference from the control, which indicates intact E-cadherin protein levels even with TGF-β treatment (Figure 3C). OLA also induced approximately a two-fold increase in mRNA expression of E-cadherin relative to the control (Figure 3B). MEO addition, however, did not significantly change the mRNA levels of E-cadherin. Low-magnification representative ICF photographs are shown in Figure A2. Next, we evaluated the functional consequences of these inhibitors on EMT by performing a wound-healing assay to evaluate migration capacity (Figure 3D,E). We observed that the percent wound closure increased, as expected, following TGF-β treatment, and did not diminish with XAV addition (Figure 3D,E), which is consistent with the E-cadherin data. Furthermore, the addition of MEO and OLA showed a modest but statistically significant decrease in wound healing relative to the control cells (Figure 3D) at 6 h, but this difference is lost at 15 h (Figure 3E). The addition of MEO, XAV, and OLA by themselves also seemed to show a small decrease in migration. Taken together, these data suggest that OLA treatment abolished EMT induced by TGF-β in NMuMG cells. In addition, no significant effects of inhibitors on cell viability were detected in a trypan blue dye exclusion test (Figure A3).

### 2.4. EMT Was at Least Partially Reversed by OLA

To determine if OLA could reverse EMT, cells exposed to a 48-h TGF-β treatment were supplemented with OLA together with fresh TGF-β. After 24 h (Figure 4A–L,a–d) or 48 h (Figure 4M–X,e–h), cells were fixed (See schedule, Figure 4a) and E-cadherin or β-catenin, vimentin and PAR intensity as well as anisotropy were quantified.

The F-actin anisotropy increased in response to TGF-β and decreased after OLA addition (Figure 4D–F,P–R,d,g). Vimentin (Figure 4G–I,c) and β-catenin also increased after EMT and decreased after OLA treatment. PAR (Figure 4S–U,e) followed the same pattern. On the other hand, E-cadherin loss in EMT was further pronounced in the presence of OLA addition (Figure 4A–C,b).

To sum up, all parameters of EMT that we measured were reversed by OLA, except E-cadherin after 24 h post-treatment.

## 3. Discussion

Our work demonstrates that untreated NMuMG cells harbor a PAR belt when they are E-cadherin rich, which is disassembled following EMT (N-cadherin rich cells). These data fit our previous work correlating E-cadherin-rich junction complexes with PAR. The function of PAR in the context of E-cadherin belts has not been elucidated until now. Thus far, the only related functional piece of evidence comes from Madin Darby Canine Kidney MDCK cells, which require TNKS activity for E-cadherin-rich epithelial belt assembly association [22]. PAR has been described (although in a nuclear context) as a molecular *glue* for multi-protein complex assembly [2]. Even though it is not the main focus of our work, we determined that nuclear and cytoplasmic PAR changed during EMT, as would be expected in a model in which F-actin changes [8,23], which argues again in favor of a relationship between PAR and the actin cytoskeleton.

Our main aim was to test the effect of PARP inhibitors in blocking or preventing EMT. We confirmed that, like in other EMT models [39], a PARP-3 inhibitor (MEO) prevented TGF-β induction of EMT, as noted by continued presence of E-cadherin through RNA and protein measurements, and by migration assays. PARP-3 is a mono-(ADP-ribosyl) polymerase. Therefore, its protective effects are not directly related to PAR abundance. 

On the other hand, we did not observe any effect of XAV939 on TGF-β induced EMT in NMuMG cells, despite the fact that TNKS inhibition by XAV939 and the subsequent modulation of WNT pathway inhibits the proliferation of small-cell lung cancer H446 cells as well as proliferation and migration of lung adenocarcinoma A549 cells [47,48]. 

Lastly, the treatment with PARP-1/2 inhibitor OLA mimicked the effect of MEO in that TGF-β induction of EMT was blocked. There was a pronounced increase in E-cadherin mRNA, which was about two-fold higher than the control. However, the protein levels were similar to the control. Importantly, similar to MEO, there was no loss of E-cadherin protein. Most importantly, both MEO and OLA demonstrated a loss of migratory capacity, which resembled the low levels seen with the control.

The role of PARP-1 and PARylation in EMT remains controversial. Musaka’s group [42,43] argued in favor of an anti-EMT role of PARP-1 and PARylation. First, they identified PARP-1 as a Smad-interacting partner in a proteomic screen. Next, using siRNA to silence PARP1/2 or through a NAD+ competitor, they further demonstrated that PARP-1 ADP-ribosylates Smad3/4 attenuated Smad-specific gene responses to TGF-β. Nevertheless, to the best of our knowledge, they were the first group reporting (see Reference [42], Appendix A) TGF-β induced nuclear PARylation in a PARP-1 dependent manner (as in NMuMG cells, Figure 1). It is unclear how PARP-1 dependent PARylation could be induced by TGF-β, and yet, be anti-EMT at the same time. 

Huang et al. (2011) demonstrate that PARP-1 is indispensable for TGF-β induced Smad3 activation in vascular smooth muscle cells. Thus, in their model, PARP-1 dependent PARylation is pro-EMT, which is similar to what we observed in NMuMG cells [41]. 

Pu et al. (2014) crossed transgenic adenocarcinoma of the mouse prostate (TRAMP) mice with PARP-1 −/− mice, and, comparing several aspects of tumor formation in wild-type, heterozygous, or PARP-1 mutated mice, concluded that impaired PARP-1 function promotes EMT and prostate tumorigenesis in vivo [49]. However, the caveat to this study was that the original PARP-1 −/− mice (Jackson Laboratories # 002779) were obtained from parental 129Sv background mice, which demonstrate a proclivity to developing testicular cancer. Thus, even with wildtype PARP-1, the mice obtained by crossing 129Sv with TRAMP mice would be expected to be more cancer-prone than TRAMP mice alone. 

NMuMG is a non-transformed mouse mammary gland epithelial cell line, which was the first to be used as a TGF-β1 induced EMT model [27]. Although these cells are epithelial and E-cadherin-rich, phenotypic variability can be observed. Even in the untreated cell culture, there are cells with pretty refractory strong epithelial junctions coexisting with more flattened and expanded cells especially in lower cell density regions. Some authors have even sub-cultured NMuMG cells generating more homogeneous clones [50]. On one hand, variability is a disadvantage in an experimental model because it requires more extensive sampling to reach significant conclusions. On the other hand, variability is one of the main characteristics of real tumors in vivo and it is also the source for tumor resistance. Furthermore, since NMuMG cells are non-transformed, they should be relatively free of epigenetic changes that might arise subsequent to neoplastic transformation, and could have influenced our experiments in other ways.

In NMuMG cells, OLA co-treatment did undoubtedly display anti-EMT effects, according to biochemical, morphological, and functional assessments. Pretreatment with OLA and MEO, but not XAV, followed by TGF-β induction of EMT in the continued presence of the inhibitors, appeared to prevent loss of E-cadherin and also inhibit migration. The PARPis and OLA co-treatment or even pre-treatment experimental schedule did not allow us to distinguish whether OLA was blocking EMT initiation, or if it was actively reversing EMT. Since this is an important question regarding an eventual extrapolation of these results to a pathological context, we next changed the experimental schedule to know if OLA could reverse EMT. An OLA 24-h post-treatment reversed EMT, according to all the measured parameters except E-cadherin, and indicates that OLA may induce its own changes favoring hybrid phenotypes (Vimentin rich, E-cadherin rich, with a morphology that tends to be epithelial) rather than just inducing the complete EMT reversal called mesenchymal to epithelial transition (MET). The existence of hybrid phenotypes has been documented before. Certain hybrid phenotypes are thought to enable the cells moving as clusters, which renders them more resistant to apoptosis than single cells [51,52]. Alternatively, since complete EMT in our cells required 48 h, maybe OLA could induce MET if given a 48-h action window. For this reason, we performed experiments with a longer exposure to OLA. According to the evaluated parameters (F-actin and β-catenin), a longer OLA treatment reverted EMT. Nevertheless, as in the case that E-cadherin was not measured, we cannot assure if there was a total reversion. 

The elucidation of the molecular mechanisms involved is beyond the scope of this work. Some speculative explanations for these results would include:OLA targeted nuclear PARP-1/2 affecting gene expressionOLA targeted PARP-1 in extracellular vesicles. PARP-1 has been detected in exosomes from colorectal cancers, melanoma, and hepatic carcinoma (EXOCARTA, Reference [53]) and in microvesicles or ectosomes (Vesiclepedia, [54]). It has been indirectly demonstrated that PARP-1 has a role in intercellular communication through exosomes [54,55]. (Vesiclepedia, [54]). Moreover, PAR in extracellular vesicles acts as a paracrine signal [56].OLA targeted PARP-3 in addition to PARP-1/2. Although OLA has been well-known in the PARP scientific community for a long time as a specific PARP-1/2 inhibitor, it has recently been demonstrated that it can also inhibit PARP-3 in vitro enzymatic activity [57]. Therefore, a new question arises as to whether OLA can inhibit PARP-3 like MEO, which explains why OLA displayed similar effects to those already who demonstrated for MEO in other systems. While we do not have the answer to this question yet, we do know that PARP-3 does not synthetize PAR. Therefore, neither PAR increase during EMT, nor PAR decrease in the presence of OLA can be explained by invoking only PARP-3. In addition, 5 µM OLA does not significantly diminish PARP-3 activity in the BT-20 breast cell line [58], which suggests that the in vivo effect of OLA on PARP-3 activity is not as evident as in vitro. OLA had non-target effects such as on the ERK (extracellular signal-regulated kinases) pathway ([46,47]) or as a substrate of permeability glycoprotein 1 (Pgp) [48,59], which is a pump that exports toxins or drugs out of cells. Pgp is overexpressed on the surface of many neoplastic cells [60] and in extracellular vesicles that mediate multi-drug resistance in prostate cancer cells [61]. 

On one hand, the synthetic lethality rationale has been the guiding light to develop a successful new therapy. On the other hand, such a rationale could bias the interpretation of the obtained results in terms of molecular mechanisms of action, with the potential practical consequence of leaving behind patients who could, otherwise, benefit from this therapy. In other words, OLA might be functioning by more than one mechanism of action and may be useful even in non-BRCA mutated cancers.

OLA effects on EMT indicate that OLA might be working by more than one mechanism of action and may be useful even in non-BRCA mutated cancers. At the same time, caution should be exercised regarding the induction of hybrid phenotypes, which could happen even in BRCA-mutated patients. The final balance may turn out to be tumor-specific, and future research should help address this issue.

## 4. Materials and Methods

### 4.1. Cell Culture and Treatments

NMuMG cells (ATCC CRL1636 TM purchased by Dr. Dhasarathy at the University of North Dakota UND, USA and submitted to Instituto de Investigaciones Biológicas Clemente Estable IIBCE, Montevideo, Uruguay, in passage 5 through a material transfer contract) were routinely cultured such as in Bhattacharya et al. (2018) [62], or in DMEM + 10% FBS (Capricorn-11A) at 37 °C and 5% CO_2_. Cell density was adjusted to achieve confluence in control wells at the end of the experiment. For example, 300,000 cells were seeded for a 10 h experiment while 30,000 were seeded for a 96 h experiment. Cells were seeded and, the following day, they were subjected to serum deprivation for 4 h to favor cell cycle synchronization [62]. Concomitant with serum replenishment, cells were exposed to 5 ng/mL TGF-β1 (Millipore-SIGMA CAT# H8541, St. Louis, MO, USA) for at least 48 h, with or without OLA (50 nM to 10 µM), XAV939 (10 µM) in DMSO or MEO328(10 µM). OLA, XAV939 and MEO328 were purchased from Tocris (Minneapolis, MN, USA). Different experimental schedules were used in order to determine whether OLA could prevent or revert the EMT induced by TGF-β (see Figure 2P, Figure 3A, Figure 4a,f). In order to check that the effect of TGF-β was specific, the cells were co-treated with SB-431542 (Millipore-SIGMA CAT#S4317, St. Louis, MO, USA) in DMSO and a vehicle control was done in parallel.

### 4.2. Specificity of the Anti-PAR Antibodies Signal in NMuMG Cells

#### 4.2.1. PAR Belt Synthesis Inhibition by XAV

Cells were detached by trypsinization and pipetting as usual and then seeded in control conditions or in the presence of 25 µM XAV [23]. Ten h later, the cells were fixed and subjected to immunocytofluorescence (ICF) using Anti-PAR antibody from ENZO (Farmingdale, NY, USA).

#### 4.2.2. PAR Digestion on Fixed Cells

The PARG effect was studied as previously [8]. The fixed cells were rinsed and incubated at RT in 15 µL PARG-reaction buffer or PARG-buff [50 mM potassium phosphate buffer pH 7.5, 50 mM KCl, 10 mM β-mercaptoethanol, 10% *v*/*v* glycerol, 1 mM DTT, and 0.1% *v*/*v* Triton-X100] for 3 h and 30 min at room temperature with or without 112 ng recombinant human PARG (Millipore-SIGMA SRP8023 lot A00634/A, St. Louis, MO, USA). After washing, the cells were subjected to ICF using Anti-PAR antibody from Becton Dickinson (Franklin Lakes, NJ, USA).

### 4.3. Immunocytofluorescence (ICF) and Confocal Microscopy

ICF was always done on 12 mm cover glasses placed in 24-well plates.The cells were rinsed in filtered PBS (fPBS, 0.22 µm pore size), fixed in 4% paraformaldehyde (PFA) in fPBS, 15 min at 4 °C, washed in fPBS, permeabilized in 0.1% Triton-X100 in fPBS, and immersed in blocking buffer (0.2% Tween, 1% BSA in fPBS) for 30 min. An indirect immunostaining procedure was performed. Additionally, the cells were incubated with the specific antibodies such as the 1:50 rabbit anti-PAR (BD551813, Becton Dickinson (Franklin Lakes, NJ, USA).), 1:50 mouse anti-PAR (ENZO BML-SA216, (Farmingdale, NY, USA), 1:50 mouse anti-VIM (ab8978, Abcam, Cambridge, MA, USA), 1:50 mouse anti-vinculin (VCL, ab18058, Abcam, Cambridge, MA, USA), and 1:100 rat anti-E-cadherin (ab11512, Abcam, Cambridge, MA, USA) diluted in blocking buffer for 2 h at 37 °C. After washing in fPBS/T (0.1% Tween), the cells were incubated (1 h, RT), with the corresponding anti-antibodies mix including goat-anti-rabbit 488 (#A-11034, Thermo Fisher Scientific, Waltham, MA, USA), goat anti-mouse 633 (#A-21052, Thermo Fisher Scientific, Waltham, MA, USA), and donkey anti-rat 488 (#A-21208, Thermo Fisher Scientific, Waltham, MA, USA) in blocking buffer for 1 h at RT. When pertinent, 1:150 phalloidin (Molecular Probes R415, Eugene, OR, USA) or Cytopainter was included in the mix to show filamentous actin (F-actin). Nuclei were counterstained with 4′,6-diamino-2-phenylindol (DAPI; 1,5 µg/mL). After washing in fPBS/T and fPBS and a final rinse in fPBS, coverslips were mounted in Prolong Gold (Molecular Probes P36930, Eugene, OR, USA) and sealed with nail polish. An OLYMPUS BX61/FV300 (Tokyo, Japan), a LEICA TCS SP5II (Wetzlar, Germany), and a ZEISS LSM800-Airyscan (Oberkochen, Germany) microscopes were used to take confocal images/stacks. All images in each experimental series were taken with the same setting at the same confocal session. If modified, all were subject to the same degree of brightness/contrast adjustment and Gaussian blur filtering, including the control without a primary antibody. The ImageJ free software (https://imagej.nih.gov/ij/download.html) was used for image processing.

### 4.4. Image Analysis

Images were analyzed using Image J software (https://imagej.nih.gov/ij/).

#### 4.4.1. Signal Intensity

The whole-field intensity measurements (RawIntDen were carried in batch (Image J commands: Process/Batch/Measure) and normalized by DAPI and the control without a primary antibody being subtracted. Data from different replicates and experiments were expressed as a percentage of the corresponding controls.

#### 4.4.2. Anisotropy

The FibrilTool ImageJ macro (http://imagej.1557.x6.nabble.com/Re-FibrilTool-ImageJ-plug-in-td5014821.html) [46] was used for anisotropy quantification carried on the three images of the stack with the highest F-actin signal after applying a Gaussian blur (once, radio:1). Statistical analyses and graphs were done using Microsoft Excel 2010. 

#### 4.4.3. Nuclear Intensity Was Quantified Using the Image J Intensity Ratio Nuclei Cytoplasm Macro

Two-tailed unequal variances Student′s t-test was applied to calculate the statistical significance. Data are represented as mean ± SEM (standard error of the mean) values.

### 4.5. RNA Isolation and quantitative realtime polymerase chain reaction (qRT-PCR)

RNA was isolated from cells using the RNeasy kit (Qiagen, Germantown, MD, USA), according to the manufacturer’s instructions. Genomic DNA was removed by on-column DNA digestion with RNase-Free DNase Set (Qiagen, Germantown, MD, USA). RNA quality and concentration was assessed using a spectrophotometer (NanoDrop, Thermo-Fisher Scientific, Waltham, MA, USA) and by electrophoresis on a 2% agarose gel. One microgram of RNA was used to synthesize cDNA using random hexamer priming and SSRT-III reverse transcriptase (Life Technologies, Carlsbad, CA, USA), which was followed by qPCR using Quantitect (Qiagen, Germantown, MD, USA) primer assays. Data were normalized against Rrn18S gene transcripts (Quantitect, Qiagen, Germantown, MD, USA). Data were derived from at least three independent biological replicates, and are represented as mean ± SEM values. Data were analyzed using the delta-delta Ct method. Statistical analyses were performed using the GraphPad Prism software, version 7.0.(https://www.graphpad.com/scientific-software/prism/).

### 4.6. Protein Isolation and Western Blots

Protein isolation and western blots were performed as in Bhattacharya et al. 2018 [62]. Briefly, proteins were extracted by lysing cell pellets in urea lysis buffer (8M urea, 1% sodium dodecyl sulfate (SDS) in Tris-HCl pH 6.5) containing CompleteTm protease inhibitors and phosphatase inhibitors (Millipore-SIGMA, St. Louis, MO, USA), and subsequent heating to 95 °C for 5 min. The protein concentration was estimated using the Qubit (Thermo-Fisher Scientific, Waltham, MA, USA) protein assay kit by following the manufacturer’s instructions. Western blots were performed and developed using the Li-COR Odyssey instrument (Li-COR Biosciences, Lincoln, NE, USA) using Luminata Forte Western HRP substrate (WBLUF0500, Millipore-SIGMA, St. Louis, MO, USA). Western blots were quantified using Image Studio Li-COR software (https://www.licor.com/bio/products/software/image_studio_lite/) and normalized to Actin. Primary antibodies used were E-cadherin, (CAT# 3195, Cell Signaling Technologies, Danvers, MA, USA) and Actin (CAT# MAB1501, Millipore-SIGMA, St. Louis, MO, USA).

### 4.7. Migration Assays

Migration assays were carried on following Lim et al. and Chan et al. [63,64]. Confluent cells in a six-well plate were serum starved for 4 h prior to treatment, and the TGF-β (for 48 h) were added to the wells prior to wounding using a sterile 200 µL tip. Three representative fields were marked and imaged immediately at time of (0 h) and a time period after (6 h or 15 h) wounding. Cell migration across the wound was analyzed using ImageJ’s MRI Wound healing tool (http://dev.mri.cnrs.fr/projects/imagej-macros/wiki/Wound_Healing_Tool). The tool measures the area of the wound, i.e., the area that does not contain cells, in each image. A ratio of the area of the wound at the start of wounding and at the end of wound closure is estimated as the percent of cell migration [65]. Data are the average of at least four independent experiments, and statistical analyses were performed using GraphPad Prism 7 software (https://www.graphpad.com/scientific-software/prism/).

## Figures and Tables

**Figure 1 ijms-20-00518-f001:**
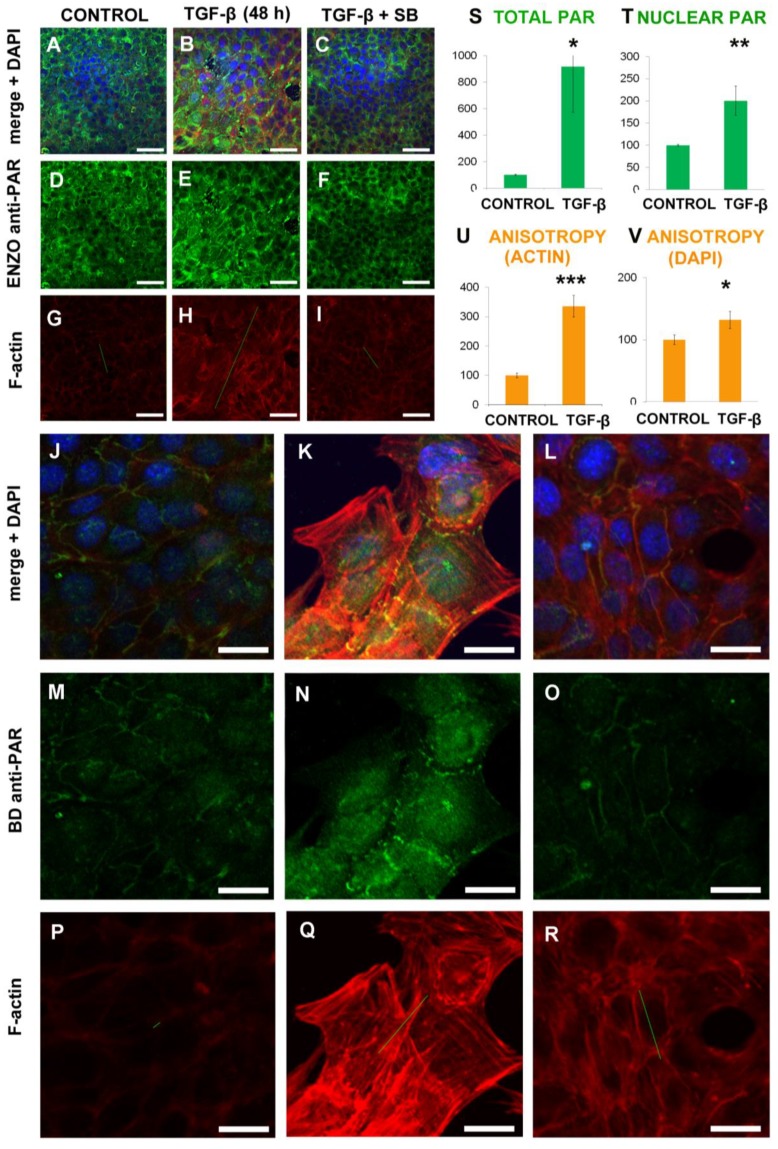
TGF-β induced NMuMG cells anisotropy increases together with changes in nuclear, cytoplasmic, and belt PAR. Control NMuMG cells were subjected just to 4 h of serum depletion. Other cells were also treated with TGF-β (5 ng/mL, 48 h) with or without 20 µM SB-431542 (SB) in 0.2% DMSO. All images are confocal single planes and have undergone a Gaussian blur with ratio = 1. Top panel (**A**–**I**). Columns correspond to treatments: control (**A**,**D**,**G**), TGF-β (**B**,**E**,**H**) and TGF-β+SB (**C**,**F**,**I**). The first raw displays merge + DAPI (blue) nuclear counterstain (**A**–**C**). PAR was detected with ENZO anti-PAR antibodies (green; **D**–**F**), F-actin, with a rhodamine-phalloidin probe (red; **G**–**I**). The green lines represent the direction and magnitude of anisotropy. Bar: 50 µm. Bottom panel: control (**J**,**M**,**P**), TGF-β (**K**,**N**,**Q**) and TGF-β+SB (**L**,**O**,**R**). The first row displays merge + DAPI (blue) nuclear counterstain (**J**–**L**). PAR was detected with BD anti-PAR antibodies (green; **M**–**O**), F-actin, with a rhodamine-phalloidin probe (red; **P**–**R**). The green lines represent the direction and magnitude of anisotropy. Bar: 10 µm (*). The graphs display control and TGF-β image quantification data. Top, green bars: total (**S**) and nuclear (**T**) PAR. Nuclear PAR signal detected with ENZO anti-PAR antibody was measured using the Intensity Ratio Nuclei cytoplasm macro. Total PAR was quantified as RawIntDen. PAR data are from at least 58 planes from seven stacks from three independent experiments. Data was normalized by DAPI (to avoid biases by variable cell density in different microscopic fields) and processed as the percentage control of the corresponding experiment. Mean ± SEM. Bottom, orange bars: anisotropy measurements carried on the F-actin (**U**) or the DAPI (**V**) channel using the FibrilTool ImageJ macro. TGF-β induced morphological changes that could be quantified through anisotropy. Anisotropy increase in F-actin images reflected the epithelial belt loss and the parallel stress fibers abundance. An anisotropy increase in DAPI images was due to nuclear shape change - from round to ovoid - and to the positioning of nuclear major axes parallel to actin stress fibers. Anisotropy data are from at least 70 planes from nine stacks from the same two independent experiments. Mean ± SEM. * *p* < 0.05, ** *p* < 0.01, *** *p* < 0.001.

**Figure 2 ijms-20-00518-f002:**
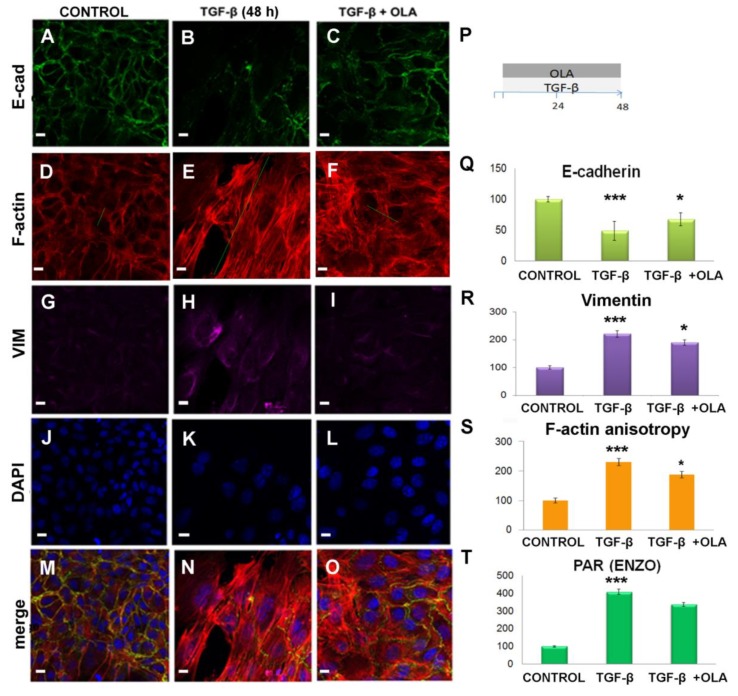
OLA prevented EMT-associated morphological changes, E-cad decrease, and VIM increase. NMuMG cells were treated with TGF-β (5 ng/mL) for 48 h in the presence or absence of 50 nM OLA and the effects were evaluated by ICF (**A**–**O**) and image quantification (**Q**–**T**). All images are confocal single planes and have undergone a Gaussian blur with ratio = 1. Columns correspond to control (**A**,**D**,**G**,**J**,**M**), TGF-β (**B**,**E**,**H**,**K**,**N**) and TGF-β + OLA (**C**,**F**,**I**,**L**,**O**) treatments. Rows display E cad (**A**–**C**), F-actin detected with phalloidin-rhodamine (**D**–**F**), VIM (**G**–**I**), DAPI counterstain (**J**–**L**), and merged channels (**M**–**O**). Bar: 10 µm. The green lines represent the direction and magnitude of F-actin anisotropy. The experimental schedule is represented in (**P)**, where the blue arrow represents time axis (h). After 4 h serum deprivation, TGF-β and OLA were added simultaneously as a continuous 48-h treatment. Not only typical EMT markers E-cad (**Q**) and VIM (**R**) but also morphological changes (**S)** were significantly affected by a low OLA concentration that showed a total PAR diminution tendency (**T**). PAR (n > 70 planes from 10 stacks) and E-cadherin (n > 150 planes from 20 stacks) data are from four independent experiments. VIM (n > 98 planes from 10 stacks) was quantified in three independent experiments. All intensity data were processed as the percentage control of the corresponding experiment and are expressed as the mean percentage control ± SEM. F-actin anisotropy data are from three independent experiments. Anisotropy was quantified on 20 control stacks and > 40 TGF-β or TGF-β + OLA stacks, using the three planes of each stack with higher F-actin signal, after subjecting them to one Gaussian blur (*r* = 1) filtering operation. Data were processed as absolute numbers and were expressed only for visual clarity as a mean percentage control ± SEM. Statistical comparisons refer to control vs TGF-β (asterisks on TGF-β bars) and to TGF-β vs TGF-β + OLA (asterisks on TGF-β + OLA bars). * *p* < 0.05, ** *p* < 0.01, *** *p* < 0.001.

**Figure 3 ijms-20-00518-f003:**
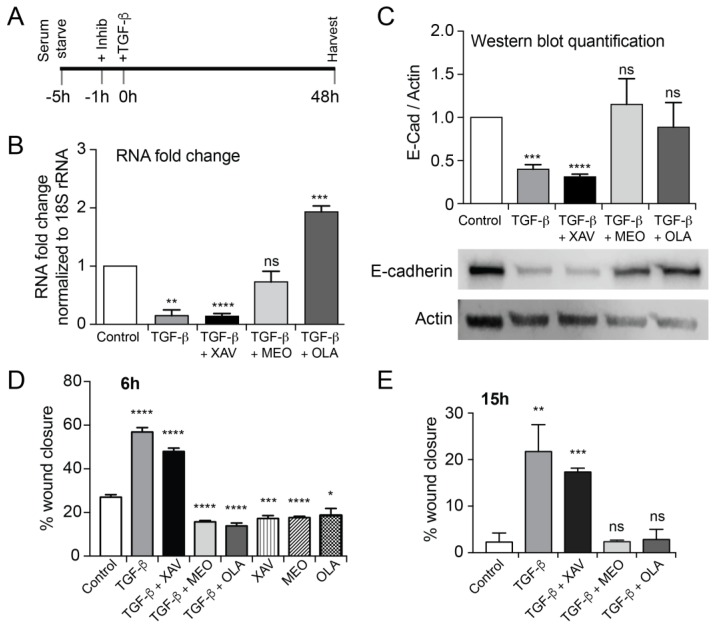
EMT is prevented by PARP-1/2 inhibitor OLA and PARP-3 inhibitor MEO but not by TNKS inhibitor XAV. In addition, 10 µM of each indicated PARP inhibitor was added, according to the experimental schedule (**A**), 1 h before treatment of NMuMG cells with TGF-β (5 ng/mL) for 48 h. E-cadherin mRNA fold increase relative to 18S rRNA (**B**) was evaluated by RT-PCR and E-cadherin protein increase relative to actin was evaluated by WB (**C**). The wound done after 48 h of TGF-β treatment, at *t* = 0 was evaluated 6 h (**D**) or 15 h later (**E**) and cell migration was measured using the “MRI wound healing” Image J plugin comparing the images at time = 0 and after 6 or 15 h of healing. Data are an average of at least three independent biological replicates and expressed as SEM. Statistical analyses (t-tests) were performed using GraphPad Prism 7 software comparing each treatment to the control. * *p* < 0.05, ** *p* < 0.01, *** *p* < 0.001, **** *p* < 0.0001, ns = not significant. See Figure A2 for ICF overview and Figure A3 for a trypan blue dye exclusion cell viability assay.

**Figure 4 ijms-20-00518-f004:**
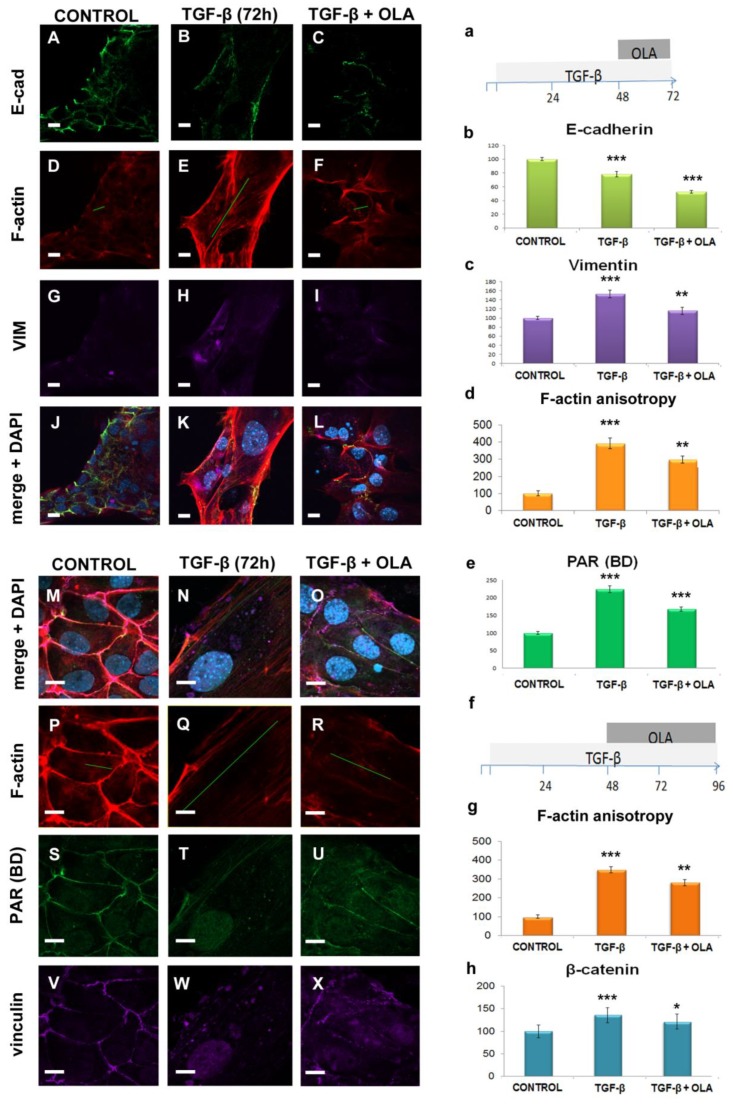
EMT was at least partially reverted by an extended OLA post-treatment. Left and top: NMuMG cells were treated with TGF-β (5 ng/mL) for 72 h in the presence or absence of 50 nM OLA during the last 24 h (schedule in **a**), and the effects were evaluated by ICF (**A**–**X**) and image quantification (**b**–**d**). ICF: Columns correspond to control (**A**,**D**,**G**,**J**; **M**,**P**,**S**,**V**), TGF-β (**B**,**E**,**H**,**K**; **N**,**Q**,**T**,**W**) and TGF-β + OLA (**C**,**F**,**I**,**L**; **O**,**R**,**U**,**X**) treatments. Rows display E cad (**A**–**C**), F-actin detected with phalloidin-rhodamine (**D**–**F & P**–**R**), VIM (**G**–**I**), merge + DAPI counterstain (**J**–**L & M**–**O**), PAR (**S**–**U**) and vinculin (**V**–**X**). Bar: 10 µm. The green lines represent the direction and magnitude of F-actin anisotropy. All images are confocal single planes and have undergone a Gaussian blur with ratio = 1. Top left panel (**A**–**L**) brightness was artificially increased by Photoshop, in a single operation per channel, to facilitate visualization without affecting intensity comparisons among different treatments. Image quantification results: E-cadherin (**b**) and VIM (**c)** were measured in parallel from the same samples of a single experiment (*n* > 50 planes from four or five stacks). F-actin anisotropy (**d**) and PAR intensity data (**e**) are from two independent experiments. Anisotropy was quantified on 10 control stacks and 20 TGF-β or TGF-β + OLA stacks. PAR (BD) was measured in 70 control and > 200 TGF-β or TGF-β + OLA planes from five or 16 stacks, respectively. Right bottom (**f**–**h**): NMuMG cells were treated with TGF-β (5 ng/mL) for 96 h in the presence or absence of 50 nM OLA during the last 48 h (schedule in **f**). Many cells were detached during this prolonged TGF-β treatment. The effects on the attached cells were evaluated by ICF and image quantification. F-actin anisotropy and β-catenin intensity data are from two independent experiments. Anisotropy was quantified on 26 control stacks and 32 TGF-β or TGF-β + OLA stacks. β-catenin was measured in > 200 planes from > 20 stacks. All intensity data were processed as the percentage control of the correspondent experiment and are expressed as the mean percentage control ± SEM. Anisotropy was quantified on the three planes of each stack with higher F-actin signal, after subjecting them to one Gaussian blur (*r* = 1) filtering operation. Anisotropy data were processed as absolute numbers and were expressed just for visual clarity as a mean percentage control ± SEM. * *p* < 0.05, ** *p* < 0.01, *** *p* < 0.001. Cell morphology and cytoskeleton alterations reflected in F-actin anisotropy measurements as well as changes in vimentin or β-catenin signals strongly suggested that OLA (which induced total PAR diminution, as expected) could revert TGF-β effects. After the short OLA post-treatment (a), E-cadherin diminution was not reverted, which raises the possibility of a partial EMT reversion.

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
