# Peer review of "PARP-1/2 Inhibitor Olaparib Prevents or Partially Reverts EMT Induced by TGF-β in NMuMG Cells"

_ijms, 2019, doi:10.3390/ijms20030518_

Reviewer 1 Report

Data is clear and manuscript is well written, but some points to be corrected.

1. Major point

In Fig 4C, it does not seem to be reduced expression of E-cadherin. I suggest to replace better one, if result is true.

2. Minor points

In Figures, some words are confusing. Please correct  (48h) in Fig2B.

And you should correct "CONTROL" and "control", "TGF-b+OLA" and "TGO".

In Fig2Q and 2R, "Q" and "R" should be separated from Y-axis number.

In Fig4b, "b" of TGF-b should be changed as symbol.

3. In line 74, change "k" of "NF-kB" to symbol.

4. In line 97, What is the meaning of "figshare" ? Please explain it in a manuscript.

5. In line 295 separate ".. PARPis .." into ".. PARP is ..".

6. In line 461, change "b" of "TGF-b" to symbol.

Author Response

We thank both reviewers for helping to correct our mistakes and for elevating the content of the manuscript. Our detailed response to the comments follows below.

1. Figure 4 has been replaced for the sake of clarity, following the reviewer’s suggestion. Please note that,  we did statistics precisely to avoid conclusions based on single images.

2. We thank the reviewer for the comment. We have made the above changes in figures 1, 2 and 4

3. This has been changed in the manuscript as noted.

4. Figshare is an online digital data repository (as explained in line 97). When the manuscript is accepted, we will provide a public link to our datasets. In the meantime, since an email is needed, we have given a private link to our datasets to the publishers.

5. Please see line 50. “PARPis” was meant as an abbreviation for “PARP inhibitors”. We have changed it to “inhibitors” on line 303 (295 in the previous version) to avoid confusion in the context of this particular sentence.

6. Thank you for catching this error. It has been changed.

Reviewer 2 Report

1. The article entitled “PARP-1/2 inhibitor Olaparib prevents or (partially) reverts EMT induced by TGF-β in NMuMG cells” submitted by Schacke et al. is an interesting work realized to investigate the impact of PARP inhibitors on the TGF-beta induced EMT. The NMuMG cell line is a model of choice to study EMT. The authors demonstrated that TGF-beta induce EMT concomitantly with an increased expression of PAR and is also associated to a PAR-belt disassembly while cells are changing in morphology to become mesenchymal. Schacke et al. then demonstrated that the TGF-beta induced EMT in this model could be counteracted by using PAR inhibitors, namely MEO and OLA but was not affected by using the XAV inhibitor. The use of Olaparib inhibitor is relevant since this compound has been approved by the FDA. Overall, the results are clear and the figures well designed but the presentation must be improved (Figure 2). The paper is well written, but some misspelling still need to be fixed (e.g. line 50 there is two points to end the sentence, line 57 “is a ring not mora than à Is a ring of not more than). The authors are clearly expert in the domain.

2. One of my biggest concern remains in the usage of only one unique model of TGF-beta induced EMT. There are multiples models (such as human HMLE cell) that could be used to validate this finding and support the concept.

3. Moreover, the usage of inhibitors such as PAR inhibitors should be substantially supported by toxicity and specificity tests and need to be presented on the paper. Do the inhibitors are impacting cell death? What are the impacts of the inhibitors on the cell proliferation?

4. The statistics of the graphs of Figure 2 and 4 need to be more detailed, the authors need to provide detailed statistics values between “control and TGFb” ; “Control and TGFb + Inhibitors” but also between “ TGFb and TGFb + Inhibitors.

5. As some minor concerns, I would remove the parenthesis on “partially” for the title and the usage of “TGO” in the figures 2&4 need to be fixed to fit with the other graphs.

Author Response

We thank both reviewers for helping to correct our mistakes and for elevating the content of the manuscript. Our detailed response to the comments follows below.

1. We thank the reviewer for their positive comments and have re-edited the paper for grammatical and typographical errors. These changes have been highlighted using the “track changes” feature in MS Word.

2. Up to now, we know what happens in EMT induced by TGF-β in NMuMG cells. We agree with you: it is worth studying this issue in other models. Nevertheless, in order to reach a more general conclusion, we should try different cell lines, possibly even from different organs. We should also apply alternative EMT inductors. Anything in-between will not allow us to arrive to a “more general”  conclusion. Unfortunately we are unable to repeat all these experiments due to lack of financial support. We plan to do these experiments as soon as we can.

3. We thank the reviewer for bringing this to our attention.

It is difficult to test the impact of these inhibitors on cell proliferation in combination with TGF-β, as TGF-β itself impairs the cell cycle. According to other groups working with breast cells, the cell cycle is not affected by OLA or MEO (5 µM) (Shariff-Askari 2018) or OLA does not affect S phase and slightly lengthens G2/M (Zhao et al 2018). Although we cannot exclude a slight cell cycle modulation, that is not the focus of our work and does not affect our conclusions regarding EMT modulation.

It has to be noted that the toxicity issue is not simple in this system, since as far as we know the chemosensitivity to most agents is different in epithelial and mesenchymal cells, decreasing after EMT. Thus, the sensitivity in untreated NMuMG cells is expected to be higher than in TGF-β treated cells. Besides, there are technical problems. For example, the typical MTT assay to test cell viability through the assesment of mitochondrial activity could be misleading in our system since (i) some cells may detach and still be alive and (ii) epithelial and mesenchymal cells have different sizes and different metabolic rates, thus making it impossible to interpret the percent of formazan absorbance in terms of percentage of living cells.

Based on our cell images, we did not notice major cellular toxicity/ cell death relative to the control treated cells.Sharif-Askari (2018) showed that in some breast cell lines (BT-20, MDA-231 and MCF-7), the IC50 of an OLA 5 day-long treatment is 10 µM while IC50 of MEO is 100 µM. According to MTT viability assays done after a 72 h treatment by Zhao et al. (2018) in CAL51, MDA-MB-231 and MCF-7 triple-negative breast cancer cells, OLA IC50 is respectively 44, 95 and 100 µM. Thus, shortening the treatment to 3 days seems to rise MCF-7 OLA IC50 to 100 µM. Our treatments were never longer than 2 days. Thus, even higher IC50s would be expected. Just in case, we didtrypan blue experiments to check in NMuMG epithelial cells that the higher working concentrations used did not affect cell viability significantly neither alone nor in combination with TGFβ (see Figure A3).

While performing a literature search to respond to the reviewer comments we realized that there are two new publications (O’Çonnor et al 2016, Sharif-Askari et al 2018) demonstrating that olaparib, which has been well-known in the PARP scientific community for a long time as a specific PARP-1/2 inhibitor , inhibits PARP-3 in vitro as well with a similar IC50. A new question arises as to whether olaparib just inhibits PARP-3 like MEO328, explaining in this simple way that it displayed similar effects to the ones already demonstrated for MEO in other systems? We do not have the answer. Nevertheless, what we know is that PARP-3 does not synthetize PAR. Therefore,  neither PAR changes in EMT nor olaparib effects (lowering PAR again) can be explained invoking just PARP-3. PARP-1/2 or tankyrases should be also involved. A tankyrase inhibitor does not block EMT, leaving PARP-1/2 as the probable main actors explaining the observed PAR changes.

Interestingly, PARP-3 activity tends to diminish but is not significantly diminished in  BT-20 cells treated with 5 µM MEO or 5 µM OLA. Accordingly, 50 nM OLA would not be expected to alter PARP-3 activity

We have always been aware that the elucidation of the molecular mechanisms involved is beyond the scope of this work (see discussion,  line 358) and that even non-PARP olaparib targets may be involved (see permeability glycoprotein 1, line 367).  Thus, your comment led us to include one more point in the discussion, refered to PARP-3 (see line 374).

4. In ICF results (Figure 2 and Figure 4), the asterisks were already refered to control vs TGFb and TGF b vs TGFb + olaparib. The comparison among control and TGF-b + inhibitors was not included. Conversely, in Figure 3 all the comparisons are relative to the control (see line 253) and we were lacking the comparison among TGF-b and TGF-b + inhibitors.

Here above is a table including all the statistical comparisons ( two-tailed t-test , unequal variances).

p-values

C vs TGF-β

TGF-β vs TGF-β+ inhibitor (which inhibitor)

C vs TGF-β+ inhibitor (which inhibitor)

Figure 1 (EMT)

Total PAR

0.021

Nuclear PAR

0.0035

F-actin ANISOTROPY

2.11 x 10 -8

DAPi ANISOTROPY

0.043

Figure 2 (OLA during)

E-cadherin

1.59x 10 -13

0.016

1.66 x 10 -5

VIM

3.31 x 10 -18

0.036

1.1 x 10 -14

F-actin anisotropy

2.11 x 10 -8

0.027

2.07 x 10 -10

PAR (ENZO)

1.32 x 10 -7

0.28

1.8 x 10 -7

Figure 3 _

Three PARPis

Ecad RNA

0.0011

0.9325 (ns)

    (XAV)

<0.0001< span="">

 (XAV)

0.0492

    (MEO)

0.2092 (ns)

 (MEO)

0.0003               (OLA)

0.0009     (OLA)

E-cad protein

0.0004

<0.0001< span="">              (XAV)

 n/a         (XAV)

0.6379 (ns)    (MEO)

 n/a       (MEO)

0.7101 (ns)       (OLA)

 n/a       (OLA)

% wound closure_6 h

<0.0001< span="">

0.0067               (XAV)

<0.0001< span="">             (XAV)

<0.0001< span="">    (MEO)

<0.0001< span="">   (MEO)

<0.0001< span="">    (OLA)

<0.0001  (OLA)

% wound closure-15 h

0.0052

0.2619    (XAV)

0.0002   (XAV)

0.0044    (MEO)

0.9517 (ns)   (MEO)

0.0061    (OLA)

0.7525 (ns)   (OLA)

Figure 4

(OLA post 24h)

E-cadherin

7.68 x 10 -11

6.66 x 10 -5

4.83 x 10 -32

VIM

3.17 x 10 -7

0.0014

0.063

F-actin anisotropy

3.01 x 10 -13

0.0055

8.14 x 10 -12

PAR (BD)

6.07 x 10 -27

9.58 x 10 -7

9.86 x 10 -19

Figure 4

(OLA post 48 h)

F-actin anisotropy

1.26 x 10 -28

0.0031

6.20 x 10 -17

Β-catenin

1.14 x 10 -10

0.015

9.25 x 10 -5

Including all these values in the figure legends can be done but we think that it would be confusing. We would suggest  including  this table as appendix 3, if you agree.

5. We have fixed the title as requested and we have standardized the treatment labels.